# Overcoming Anxiety Disorder by Probiotic *Lactiplantibacillus plantarum LZU-J-TSL6* through Regulating Intestinal Homeostasis

**DOI:** 10.3390/foods11223596

**Published:** 2022-11-11

**Authors:** Guanlan Liu, Israr Khan, Yuxi Li, Yun Yang, Xuerui Lu, Yafei Wang, Junxiang Li, Chunjiang Zhang

**Affiliations:** 1School of Life Sciences, Lanzhou University, Lanzhou 730000, China; 2Key Laboratory of Cell Activities and Stress Adaptations, Ministry of Education, Lanzhou University, Lanzhou 730000, China; 3Gansu Key Laboratory of Biomonitoring and Bioremediation for Environmental Pollution, Lanzhou University, Lanzhou 730000, China; 4School of Pharmacy, Lanzhou University, Lanzhou 730000, China; 5Gansu Key Laboratory of Functional Genomics and Molecular Diagnosis, Lanzhou University, Lanzhou 730000, China

**Keywords:** anxiety, GABA, gut microbiota, intestinal mucosal barrier, *Lactiplantibacillus plantarum*

## Abstract

*Lactiplantibacillus plantarum LZU-J-TSL6* with high γ-aminobutyric acid (GABA) production (3.838 g/L) was screened and isolated from the Chinese fermented food snack “Jiangshui”. The improvement effect on anxiety disorder was explored using mice as animal models. In vitro results revealed that *LZU-J-TSL6* had the potential to colonize the intestine (*p* < 0.01) and the anxiety-like behavior of the mice after seven days’ gavage with *LZU-J-TSL6* was significantly improved (*p* < 0.01) when compared to the model group. *LZU-J-TSL6* was able to effectively increase the GABA content in the mice hippocampus (*p* < 0.0001) and restore some markers related to anxiety such as brain-derived neurotrophic factor (BDNF), glial fibrillary acidic protein (GFAP), and 5-hydroxytryptamine (5-HT). Simultaneously, it had a certain repair effect on Nissl bodies and colon tissue in mice hippocampus. In addition, *LZU-J-TSL6* increased the relative abundance of beneficial bacteria *Bacteroides* and *Muribaculum*, thereby regulating the imbalance of intestinal microbiota caused by anxiety disorder. It also affects the nerve pathway and intestinal mucosal barrier by increasing the content of glutamine and γ-aminobutyric acid and other related metabolites, thereby improving anxiety. Therefore, the GABA-producing Lactobacillus plantus LZU-J-TSL6 can be used as a probiotic to exert an indirect or direct anti-anxiety effect by maintaining the balance of the intestinal environment, producing related metabolites that affect nerve pathways and repair the intestinal mucosal barrier. It can be used as an adjuvant treatment to improve anxiety disorders.

## 1. Introduction

Anxiety disorder is a common clinical affective disorder characterized by widespread and persistent anxiety or repeated attacks of panic, accompanied by autonomic nervous dysfunction and abnormal behavior. New evidence suggests that anxiety is becoming a global problem and that the prevalence of the disorder is rising rapidly in developing countries [1,2].

In China, with a lifetime prevalence of 7.6%, it has become the most common mental disorder [3]. Since the prevalence of anxiety disorder is relatively high it may interfere with the ability to cope with daily life events. Therefore, in an excessively severe or frequent state, it tends to be chronic and substantial coinfection and is known as pathological [4,5]. 

Both physical and mental symptoms of chronic development in anxiety patients may lead to decreased ability, altered body condition, social and occupational impairment, heavy economic burden, and poor quality of life [6]. It is essential to distinguish between different states of anxiety, such as episodic, transitory, or trait anxiety. The symptoms of anxiety might manifest in response to environmental cues, as in phobias, or they can be asymptomatic. Moreover, the manifestation of anxiety state changes with time as new challenges or cognitive strategies are put in place, or respond to changes in surroundings.

In modern medicine, changes in neurotransmitters are considered the essential neurobiochemical mechanism for the occurrence of anxiety disorder. The γ-aminobutyric acid (GABA) is an inhibitory neurotransmitter that is distributed in the central nervous system (CNS) and has a universal inhibitory effect on neurons in the nervous system. In addition, all levels of GABA in the midbrain from the limbic structure to the cortex can inhibit excitatory conduction of 5-HT (5-hydroxytryptamine) and norepinephrine (NE) pathways [7]. There is an interaction between GABA and 5-HT. 5-HT regulates the stability of GABA neurons, and GABA receptor agonists can inhibit the expression of 5-HT in the hippocampus. Thus, GABA receptor antagonists should exert anti-anxiety effects through the 5-HT system [8]. GABA was able to regulate DAergic neurons through GABA receptors, thus playing an anti-anxiety role [9,10].

Although selective serotonin reuptake inhibitors (SSRIs) and cognitive-behavioral therapy (CBT) are the first-line treatment options for anxiety disorder, the clinical treatment for anxiety disorder is still insufficient, and the proportion of patients receiving appropriate treatment is low [11]. In addition, even if patients with anxiety receive enough first-line treatment, at least one-third of them still suffer from adverse reactions during the treatment [12].

Long-term medication can have negative side effects; thus, an increasing number of patients prefer to treat anxiety disorders with complementary and alternative therapies (such as relaxation techniques, nutritional supplements, massage, and acupuncture) [12]. Clinical investigations have reported that certain probiotic strains, in particular, *Bifidobacterium* and *Lactobacillus*, can help relieve stress, anxiety, and depressive symptoms [13,14,15]. Over the last decade, there has been an increase in the number of studies that have used probiotics to alter CNS function. Probiotics can influence psychology and behavior by affecting brain-derived neurotrophic factors (BDNF), GABA, 5-HT, and dopamine (DA) [16,17,18,19,20]. 

It has been reported that there are many potential GABA-producing or metabolic bacteria (such as *Bacteroides* and *Parabacteroides*) in the intestinal tract [21], that can regulate disorders related to depression, anxiety, and other mental health problems. However, the isolation of GABA-producing bacteria from human intestinal microflora required specific culture conditions. As a beneficial microorganism of food safety grade (GRAS), lactic acid bacteria can be widely colonized in humans, animals, and plants, producing a wide range of probiotic factors and exerting beneficial effects [22]. Although lactic acid bacteria have been reported to be used in the production of GABA and as microecological modulators, they have become alternative therapies for the treatment of certain neurological and neurodegenerative diseases [23,24]. However, previous studies have not combined the fact that GABA is the most important inhibitory neurotransmitter in anxiety disorder to screen probiotics for improving anxiety disorder. 

Fermented food has always been a major source of isolated lactic acid bacteria, and Jiangshui is very popular as one of the characteristic fermented snacks in Northwest China. In addition, some researchers found that lactic acid bacteria were the dominant species in the microbial composition of Jiangshui [25]. Therefore, Jiangshui can be used as one of the sources for the isolation of lactic acid bacteria. Early studies on the improvement of anxiety by probiotics (such as *Bifidobacterium lactis*, *Lactobacillus acidophilus*, and *Lactobacillus bulgaricus*) were mainly focused on the repair effect of probiotics on the intestinal microecological balance [21,26,27]. Therefore, in this study, we propose to screen a *Lactiplantibacillus plantarum* strain isolated from the Chinese special fermented food snack Jiangshui with high GABA production. The efficacy of GABA supplementation through probiotic therapy in alleviating anxiety disorder in mice was explored while paying attention to the regulation of intestinal microecology balance by probiotics.

## 2. Materials and Methods 

### 2.1. Screening Strains with Relatively High GABA Yield

Seven GABA-producing strains were screened from 288 strains independently isolated in the laboratory (The sources of isolates are detailed in Table 1): *GG-8-3A, LX-1-3, LX-4-7, LZU-J-TSL6, MG*(*MY*)*-7B, QX*(*A*)*-8,* and *YBT-1-2,* and further strains were screen-out with relatively high GABA production. Seven strains were inoculated into MRS medium (10 g of peptone, 10 g of beef paste, 5 g of yeast extract powder, 20 g of glucose, 5 g of sodium acetate, 1 mL of Tween-80, 2 g of diammonium citrate, 2 g of dipotassium hydrogen phosphate, 0.58 g of magnesium sulfate heptahydrate, and 0.25 g of manganese sulfate monohydrate, dissolved in 1 L of distilled water) and cultivated at 37 °C for 16 h as fermentation seeds; then 4% of the inoculum was put into a liquid culture containing Glucose-Yeast Extract-Peptone (GYP) Medium, and cultured at 37 °C for 48 h. Subsequently, the solution was centrifuged at 2000 rpm for 10 min, and the supernatant was stored at 4 °C for use. A standard curve was made by using GABA with a concentration of 1 mg/mL as the standard stock solution. 

The GABA-producing ability of each strain was quantitatively determined by high-performance liquid chromatography [28]. Using a Waters-C18 column (250 × 4.6 mm, 5 μm) with acetic acid-sodium acetate buffer (pH 5.80): methanol 55:45 as the mobile phase, 20 μL was injected manually at a flow rate of 0.8 mL/min, a column temperature of 30 °C and a detection wavelength of 334 nm. The main reagents included: GABA standard solution, derivatization reagent o-phthalaldehyde (OPA), potassium dihydrogen phosphate buffer, and acetic acid-sodium acetate buffer salt solution (15 mmol/mL). Six hundred microliters of OPA and eight hundred microliters of potassium phosphate buffer were successively added into two hundred microliters of the standard sample or fermentation broth, and the samples were injected within 2 min of being filtered through a 0.22 μm membrane filter once or twice, and then the chromatographic column was flushed for 2 h.

### 2.2. 16S rRNA GENE Sequencing and Phylogenetic Analysis of Strain LZU-J-TSL6

The bacterial suspension from the culture of strain *LSU-J-TSL6* was submitted to a commercial testing company as a sample for the subsequent sequencing process. A TIANamp Bacteria DNA Kit (TIANGEN Biotech (Beijing) Co., Ltd., Beijing, China) was used to obtain high-quality genomic DNA of the strains. The 16S rDNA was amplified as described by Sibley et al. [29] The 16S rRNA gene sequencing of the strain was performed by Beijing qingke biotechnology co. Ltd. and identified using the BLAST engine (NCBI) [30]. The phylogenetic tree was made using MEGA (molecular evolutionary genetic analysis, version 6.0, Auckland, New Zealand) software [31]. The strain *LSU-J-TSL6* nucleic acid sequence is shown in Appendix A.

### 2.3. Scanning Electron Microscope

First 1 mL of the bacterial suspension was transferred to a 1.5 mL EP tube, the tubes were centrifuged at 3000 rpm for 3 min, and the bacterial precipitate was washed thrice with normal saline. Then, 1 mL of 2.5% glutaraldehyde (2 mL) was added to the bacterial precipitate, thoroughly mixed, and fixed at 4 °C for 12 h. After centrifugation at 3000 rpm and 3 min, 2.5% glutaraldehyde was pipetted out, and the bacterial precipitate was washed with normal saline and diluted 10 times. A 20 μL sample of the diluted bacterial liquid was pipetted out and uniformly coated on a clean glass slide, air-dried, and then, observed under a scanning electron microscope [32].

### 2.4. In Vitro Determination of Acid and Choline Tolerance of Strains

A 1 mL sample of the bacterial suspension was inoculated into 9 mL artificial gastric juice (pH = 3.0), cultured at 37 °C for 3 h at 90 rpm, and then 1 mL was added into 9 mL artificial intestinal juice (pH = 9.0) and cultured at 37 °C for 9 h at 90 rpm. At 0, 3, 6, 9, and 12 h in the experimental process, 0.1 mL bacterial solution was diluted 10 times to 10^–7^ to 10^–9^ according to the 10-fold dilution method, and spread on a culture dish containing MRS solid medium and incubated at 37 °C for 36 h. The number of colonies was counted and the survival rate was calculated according to the following formula [33]:$$\mathrm{Livability}\;=\frac{\;\mathrm{Number}\;\mathrm{of}\;\mathrm{colonies}\;\mathrm{in}\;a\;\mathrm{certain}\;\mathrm{period}-\mathrm{Initial}\;\mathrm{colony}\;\mathrm{number}}{\;\mathrm{Initial}\;\mathrm{colony}\;\mathrm{number}}\times 100\%$$

### 2.5. Animals and Treatments

Thirty-five SPF grade C57/bl6 male mice (6 weeks of age and weighing 16–18 g) were purchased from Lanzhou Veterinary Institute, Chinese Academy of Agricultural Sciences. Prior to the experiment, the mice underwent a one-week acclimation period at a temperature of 22 ± 2 °C, relative humidity of 40–55%, and a 12 h light/dark cycle alternating with a free diet (normal mice feed and sterile water). During the experiment, the mice were randomly divided into five groups, and seven mice per cage were placed under the same conditions for the experiment. The feeding and use of experimental animals in this study were approved by the Animal Ethics Committee of Lanzhou University (EAF2021040) and followed the relevant ethical regulations.

In order to induce anxiety in the mice, restraint stress modeling was used. Four groups of mice (model group, GABA group, *Lactiplantibacillus plantarum LZU-J-TSL6* group, and alprazolam group) other than the normal group were modeled. After the adaptation period, the mice were put into a 50 mL centrifuge tube with their heads facing the bottom, and the tube was placed horizontally. The mice were constrained for 2 h on the first day, and then this was raised in 2 h steps to 8 h per day, after which the mice were constrained for 8 h each day during the two weeks [34]. After the modeling period, mice in the GABA group, *Lactiplantibacillus plantarum LZU-J-TSL6* group, and alprazolam (prescription drug) group were treated with GABA (0.3 mg/0.2 mL/day), *Lactiplantibacillus plantarum LZU-J-TSL6* (4 × 10^8^ CFU/kg/day), and alprazolam (0.0035 mg/0.2 mL/day) suspended in normal saline. The treatment period was seven days. 

During the whole raising period of the mice, drinking water was changed every two days, food was changed once a day, and mice padding was changed twice a week.

### 2.6. Behavioural Testing 

On the 14th day (the end of the modeling period) and the 21st day (the end of the treatment period), the mice were tested for behavior (*n* = 4–6). After the behavior test of each group of mice, the number of times and the residence time (s) in the center of the playground were used as indicators to evaluate the behavior of the mice. The open-field test is a measure of anxiety behavior [35], and the mice autonomous activity analyzer used was purchased from Beijing Zhimo Duobao Biotechnology Co., Ltd. (Beijing, China).

### 2.7. Enzyme-Linked Immunosorbent Assay of Hippocampus and Serum in Mice 

#### 2.7.1. Blood Collection and Enzyme-Linked Immunosorbent Assay

Mice blood was collected by harvesting eyeballs before the mice were euthanized. The blood sample was kept at room temperature until the blood naturally coagulated, and centrifuged for 10 min at 4000 rpm. The serum was collected to determine the content of 5-HT in the mice according to the instructions of the mice 5-HT ELISA kit.

#### 2.7.2. Collection of Mice Hippocampus and Enzyme-Linked Immunosorbent Assay

After the mice were euthanized by cervical dislocation, the whole brain was stripped. Then the hippocampus was extracted on ice and immediately frozen in liquid nitrogen for further use. The hippocampus was manually ground with a tissue grinder (Tiangen third-generation high-speed tissue grinder), and then the contents of γ-aminobutyric acid (GABA), brain-derived neurotrophic factor (BDNF), and glial fibrillary acidic protein (GFAP) in the hippocampus of the mice were detected according to the instructions of ELISA kit [29]. All kits were purchased from the Jiancheng Institute of Bioengineering, Nanjing, China.

### 2.8. Slice Staining of Mice Colon Tissue and Hippocampus Tissue

#### 2.8.1. Hematoxylin-Eosin (HE) Staining of Mice Colon Tissue

A 1-cm length of the distal mice colon tissue was cut out, and the sample was immediately fixed with 10% neutral paraformaldehyde solution. After being embedded in paraffin, the sample was made into tissue blocks, and cut into paraffin sections with a thickness of 5 μm, and then the tissues were stained with HE. The pathological damage of the colon tissue was observed under an optical microscope.

#### 2.8.2. Nissl Corpuscle Staining in Hippocampus of Mice

The fresh brain tissue of the mice was fixed in 10% neutral paraformaldehyde solution, dehydrated, paraffin-embedded, and made into tissue blocks, then cut into paraffin sections with a thickness of 5 μm. After being stained with Nissl Stain (toluidine blue), the pathological damage of Nissl bodies in the hippocampus of the mice was observed under an optical microscope.

### 2.9. Determination of Nrf-2 and ZO-1 mRNA Expression in Colonic Tissues

#### 2.9.1. RNA Extraction and Detection of Concentration and Purity

Thirty milligrams of tissue were thoroughly ground in liquid nitrogen. After collection, 1 mL TRizol was added to each sample, which was then vortexed and mixed, and left at room temperature for 5 min. Then the RNA was extracted, and the concentration and purity were detected as follows: First, 1 mL Trizol was added, vortexed, and mixed, and then transferred to a 1.5 mL EP tube for 5 min at room temperature. Then 0.2 mL chloroform was added to the EP tube, shaken vigorously for 15 s, left at room temperature for 2 min, and centrifuged at 12,000× *g* and 4 °C for 15 min. A 0.4 mL sample of the supernatant was pipetted and 0.4 mL of isopropanol was gently mixed into the EP tube. After standing at room temperature for 10 min, the supernatant was centrifuged at 12,000× *g* and 4 °C for 10 min. A white precipitate could be clearly seen after the supernatant was discarded. Subsequently, 1 mL of 75% ethanol was added (75% ethanol was prepared with DEPC water and 100% absolute ethanol), and the precipitate was gently washed and centrifuged at 8000× *g* and 4 °C for 5 min. The supernatant was then discarded, and the residual ethanol was removed using a pipette. Finally, the ethanol was volatilized by opening the lid on ice for 5 min and dissolved by adding 30 μL of DEPC H2O to obtain RNA. Subsequent RNA reading and purity testing were performed by Shanghai Baiqu Biomedical Technology Co., Ltd. (Shanghai, China).

#### 2.9.2. Reverse Transcription of RNA

The reaction mixture was prepared on ice according to Appendix A. In order to ensure the accuracy of the reaction mixture, the Master mix was prepared according to the amount of reaction number +2, and then the calculated amount of RNA was added to each reaction tube. The mixture was mixed and put into a PCR instrument, and the reaction was carried out at 42 °C for 2 min before being stored in a 4 °C refrigerator until the next step.

The reaction mixture was then prepared on ice according to Appendix A. In order to ensure the accuracy of the reaction mixture, the Master mix was prepared according to the amount of reaction number +2, and then 10 μL was filled into the reaction tube of the previous step. The reaction was carried out at 37 °C for 15 min and then at 85 °C for 5 s. The sample was then stored in a refrigerator at 4 °C until next used.

#### 2.9.3. PCR Detection and Data Analysis and Calculation

Primer sequences were designed, and Q-PCR was performed. The primer sequences and PCR reaction systems are shown in Appendix A, respectively.

The experimental data analysis and calculation were as follows: CtA1 was set as the Ct value of the target gene of sample 1, CtB1 was set as the Ct value of the reference gene of sample 1, CtA2 was set as the Ct value of the target gene of sample 2, and CtB2 was set as the Ct value of the reference gene of sample 2, then the target gene expression levels of sample 1 and sample 2 could be approximately calculated as (2-DDCt): ddCt = (CtA2 − CtB2) − (CtA1 − CtB1) = X. The above experimental procedures and data analysis and calculation were completed by Shanghai Baiqu Biomedical Technology Co., Ltd. (Shanghai, China).

### 2.10. Sequencing of Intestinal Flora and Metabolomic Detection and Analysis

#### 2.10.1. Extraction and PCR Amplification of Genomic DNA from Intestinal Microbiota

The genomic DNA of the fecal samples was extracted according to the manufacturer’s instructions using the DNA strong extraction kit produced by Bio Corp. (Salem, MA, USA), and then the purity and concentration of DNA were detected by agarose gel electrophoresis. An appropriate amount of sample DNA was placed in a centrifuge tube and the samples were diluted to 1 ng/μL with sterile water. Based on the selection of sequencing region, PCR was performed using the diluted genomic DNA as a template using a specific primer with a Barcode. Phusion High-Fidelity PCR Mastermix with GC Buffer from New England Biolabs and high-performance hi-fi enzymes were used throughout the process to ensure amplification efficiency and accuracy [36]. The PCR expansion and high-throughput sequencing of the samples, and the amplification primers designed with the variable sequence of the 16S rRNA V3-V4 region as the target are shown in Table 2.

#### 2.10.2. Mixing and Purification of PCR Products

The PCR mixture (25 μL) contained 1× PCR buffer, 1.5 mM magnesium chloride, 0.4 μM deoxynucleoside triphosphate, 1.0 μM of each primer, and 0.5 U Ex Taq (TaKaRa, Dalian), and 10 ng of soil genomic DNA. The PCR amplification protocol consisted of an initial denaturation at 94 °C for 3 min followed by 30 cycles of 94 °C for 40 s, 56 °C for 60 s, 72 °C for 60 s, and a final extension at 72 °C for 10 min. Two PCR reactions were performed for each sample and combined after PCR amplification [37]. DNA concentration and purity were determined by NanoDrop (Model: Thermo Fisher 2000, Instrument manufacturer: Thermo Fisher, USA). PCR products were examined by electrophoresis on a 1% agarose gel. The product was recovered from the target band using a gel recovery kit provided by Qiagen, and the concentration and quality of the product were determined by Nanodrop.

#### 2.10.3. Library Construction

A Truseq DNA PCR-free sample preparation kit was used to construct the library, and the constructed library was quantified by Qubit and Q-PCR. 

#### 2.10.4. Computer Sequencing and Data Processing

After the library passed the test, sequencing was performed using NovaSeq6000. The sequencing and data processing were performed by Shanghai Baiqu Biomedical Technology Co., Ltd. 

#### 2.10.5. Extraction of Metabolites

A 50 mg sample of stool was accurately weighed, to which 1 mL of extraction solution (VAcetonitrile: VAhol: VAwater = 2:2:1) was added. The extraction solution also contained an isotope label internal standard mixture. The samples were ground at 35 Hz for 4 min and sonicated in an ice-water bath for 5 min. This process was repeated 2–3 times. The samples were then cooled to −40 °C and left to stand for 1 h before being centrifuged at 4 °C and 12,000 rpm (centrifugal force 13,800× *g*, radius 8.6 cm) for 15 min. The centrifuged supernatant was injected into a sample injection bottle for sample injection. The sequencing and data processing were performed by Shanghai Baiqu Biomedical Technology Co., Ltd.

### 2.11. Bioinformatics and Statistical Analysis 

All effective tags of all samples were clustered using the Uparse algorithm (Uparse v7.0.1001, http://www.drive5.com/uparse/ (accessed on 10 March 2021)) [38], and species annotation analysis was carried out using the Mothur method and SSUrRNA database [39] of Silva 138 (http://www.arb-Silva.de/ (accessed on 10 March 2021)) [40] (with a threshold of 0.8). The phylogenetic relationships of all OTU representative sequences were obtained by fast multi-sequence alignment using Muscle software [41] (Version 3.8.31, EMBL-EBI, Cambridgeshire, UK, http://www.drive5.com/muscle/ (accessed on 10 March 2021)). The Observed-OTUs, Chao1, Shannon, Simpson, ace, Goods-coverage, and PD_whole_tree index were calculated by Qiime software (Version 1.9.1, http://qiime.org/ (accessed on 10 March 2021)), and the dilution curve, Rank abundance curve, and species accumulation curve were using R (Version 2.15.3, Windows, Redmond, DC, USA). The difference between groups in terms of the Alpha diversity index was analyzed by R. Unifrac distance was calculated using Qiime software (Version 1.9.1), and the UPGMA sample cluster tree was constructed. R was also used to draw PCA, PCoA, and NMDS diagrams. The above analysis is handled by Shanghai Baiqu Biomedical Technology Co., Ltd., Shanghai, China.

## 3. Results 

### 3.1. Screening of GABA-Producing Strains and Evaluation of Their Characteristics

HPLC results showed that, among the seven GABA-producing strains, the *LZU-J-TSL6* strain produced the most GABA (3.838 g/L) (Figure 1A). NCBI BLAST database results revealed that the *LZU-J-TSL6* strain was closely related to *Lactobacillus plantarum ML1* (Figure 1B) and scanning electron microscopy also confirmed that *LZU-J-TSL6* had a similar morphology to *Lactiplantibacillus plantarum* (Figure 1C). Compared with *Lactobacillus rhamnosus GG (LGG)*, *LZU-J-TSL6* exhibited a higher survival rate of 76% in simulated gastric fluid (Figure 1D). Similar results were observed in simulated intestinal fluid, where the survival rate of *LZU-J-TSL6* was ultimately higher than that of *LGG* (Figure 1E).

### 3.2. Effect of LZU-J-TSL6 on the Behavior of Anxiety Mice

To determine the improvement effect of *LZU-J-TSL6* on the behavior of anxiety mice, an anxiety mice model was constructed using the chronic restraint stress method and intra-gastrically administered with *LZU-J-TSL6* strain (Figure 2A). Fourteen days after modeling, the results of a mice autonomous activity behavior analysis revealed that the normal group of mice after restraint stress modeling, the number of mice entering the center of the active field (Figure 2B), and the time to stay in the center (Figure 2C) were all significantly reduced (*p* < 0.01) compared to the normal group. However, after the intervention with *Lactiplantibacillus plantarum LZU-J-TSL6*, the number of mice entries into the center of the active field (Figure 2D) and the time to stay in the center (Figure 2E) returned to almost the same level as in the normal group. The alprazolam-treated mice group showed little modification of their behavior. 

### 3.3. Effect of LZU-J-TSL6 on the Levels of Related Factors in Hippocampus and Serum of Anxiety Mice

In order to explore the effect of *LZU-J-TSL6* on anxiety disorder, an enzyme-linked immunosorbent assay was used to determine the related factor level. The administration of *Lactiplantibacillus plantarum LZU-J-TSL6* increased the GABA level in the hippocampus of the mice significantly (*p* < 0.0001) compared with the model group (Figure 3A), while it was second to the GABA group. The hippocampal BDNF content was also significantly increased in the *LZU-J-TSL6* and GABA groups (*p* < 0.001) (Figure 3B), with *LZU-J-TSL6* being slightly higher than the GABA group. Hippocampal glial fibrillary acidic protein (GFAP) content was significantly reduced in the *LZU-J-TSL6* and GABA groups (*p* < 0.001) (Figure 3C). Compared with the model group, the serum 5-hydroxytryptamine (5-HT) level of the mice was also positively improved after the intervention with *LZU-J-TSL6* and GABA (*p* < 0.01) (Figure 3D).

### 3.4. Repairing Activity of LZU-J-TSL6 in Intestinal Environment and Hippocampus

To explore the effect of anxiety on the intestinal environment, the expression levels of the Nrf-2 gene and ZO-1 gene in colonic tissues were detected. It was found that after treatment with *Lactiplantibacillus plantarum LZU-J-TSL6*, the expression level of the Nrf-2 gene in the colon of mice was significantly increased (*p* < 0.05) compared with the model group. However, the expression level of the Nrf-2 gene in the colon of the GABA group was not improved and was lower than that of the model group (Figure 4A). The expression level of the ZO-1 gene in the mice colon after treatment with *Lactiplantibacillus plantarum LZU-J-TSL6* was significantly higher than the model group (*p* < 0.01). The results of colonic ZO-1 gene expression level in the GABA group were consistent with the NrF-2 gene expression level (Figure 4B).

An investigation of the influence of restraint stress on colon tissue structure compared with the blank control group showed that the muscle layer structure of colon mucosa in the model group was disrupted (Figure 4C). The number of goblet cells decreased, and the crypt structure was disordered. However, after being treated by Lactiplantibacillus plantarum LZU-J-TSL6, the structure of the mucosal muscle layer was relatively complete, the number of goblet cells increased, and the crypt structure was arranged and compact. The improvement of colon tissue in the GABA group was less than that in the LZU-J-TSL6 group.

The results of Nissl staining showed that the cells in the dentate gyrus (DG) of the hippocampus in the model group were sparse and loosely arranged (Figure 4D). This phenomenon was improved in the *LZU-J-TSL6* and GABA groups. In the *LZU-J-TSL6* group, the cells in the dentate gyrus (DG) of the hippocampus were dense and arranged normally, and the effect was better than that in the GABA group. There was no significant difference in Nissl corpuscles in the hippocampal CA1 region of the mice in each group (Figure 4D). 

### 3.5. Effects of LZU-J-TSL6 Treatment on Intestinal Flora Function and Microbial Diversity in Restraint Stress Model Mice

To determine the effects of restraint stress modeling and treatment with *LZU-J-TSL6* on the diversity and abundance of intestinal microbiota, 16S rRNA was analyzed by Miseq sequencing. The abundance (Shannon index) of the microbial community in the model group was higher than that in the normal, GABA, and *LZU-J-TSL6* groups (Figure 5A). The abundance of the model group after treatment with *LZU-J-TSL6* was consistent with the blank group, while the abundance of the GABA group was very low. In addition, the results of principal coordinate analysis (Figure 5B) showed that the phylogenetic community of the restraint stress modeling group was significantly different from that of other groups, and the model group was separated from other groups. These results suggested that restraint stress modeling altered the composition of intestinal flora, and *LZU-J-TSL6* reduced the impact of modeling on the changes in the intestinal flora. Venn diagram (Figure 5C) analysis also showed that modeling significantly affected the structure of the intestinal flora.

The relative abundance at the phylum level was analyzed for four groups of samples (Figure 5D). The dominant phyla in the intestinal microflora were Bacteroidetes and Scleroderma (relative abundance > 15%). The relative abundance of Bacteroidota was decreased and Firmicutes was increased in the model group when compared with the control group. Treatment with traditional Chinese medicine can reduce mycetoma Chlamydomonas and ease the Bacteroides, suggesting that it is related to the relief of mental disorders [42]. In addition, a large number of Scleroderma are associated with mental diseases, such as Alzheimer’s disease and autism [43,44]. After the LZU-J-TSL6 intervention, the abundance of Bacteroidota and Firmicutes was adjusted to approximately that of the blank group (Figure 5E,F). At the genus level (Figure 6A), the results show an increase in the relative abundance of Helicobacter (Figure 6E), Dubosiella (Figure 6D), and Coriobacillae_UCG-002 (Figure 6C) after stress modeling. The relative abundances of Bacteroides (Figure 6B) and Muribaculum (Figure 6F) (*p* < 0.01) were significantly increased after administration of LZU-J-TSL6, while those of Helicobacter, Dubosiella, and Coriobacillae_UCG-002 (*p* < 0.01) were decreased. 

In a few published prospective human studies evaluating microbiota and neurodevelopment, *Bacteroides* showed a correlation between higher relative abundance and neurodevelopment [45]. The abundance of *Muribaculum* was positively correlated with cognitive function [46]. *Helicobacter* infection is associated with anxiety and depression and is linked through the brain-gut axis, serving as a bridge. In other words, anxiety and depression are more likely to occur after *Helicobacter* infection, and anxiety and depression will reversely affect people infected with *Helicobacter* [47,48,49]. Flavonoid treatment can reverse the imbalance of intestinal microorganisms by reducing the abundance of *Dubosiella* in intestinal microorganisms [50]. The relative abundance of *Coriobacillaceae_UCG-002* was decreased, which plays a vital role in improving intestinal inflammation and temporarily protects the large intestine of mammals [51]. It was deduced that *LZU-J-TSL6* had a positive regulatory effect on the intestinal flora.

### 3.6. Prediction of Metabolic Pathway and Changes of Related Differential Metabolites in Mice before and after Intervention with LZU-J-TSL6

To explore the changes of related metabolites after intervention with *LZU-J-TSL6*, differential metabolite analysis, and metabolic pathway prediction were conducted. A total of 18 differential metabolites were screened out in the positive and negative ion modes (Figure 7A,B). In particular, the metabolic contents of D-glutamine and GABA were significantly increased after the intervention with *LZU-J-TSL6*, and the increase of GABA was particularly significant. The relative contents of other metabolites such as taurine, histamine, dopamine, glutamic acid, pyrrolidinoic acid, L-phenylalanine, and L-serine were increased to a certain extent. The amounts of adenosine, tryptamine, cannabinoid, morphine, melatonin, and serotonin before and after treatment were not significantly different. The significantly different metabolites were introduced into KEGG (Kyoto Encyclopedia of Genes and Genomes) for metabolic pathway analysis to identify the main signaling pathways and biological metabolic pathways in which they participated. An analysis of the KEGG metabolic pathways involved in differential metabolites before and after the intervention with *LZU-J-TSL6*–the metabolic pathway for neuroactive ligand-receptor interaction–is shown in Figure 8.

## 4. Discussion

The prevalence of anxiety has an extremely profoundly negative impact on the patient’s physical capacity, economic burden, and quality of life. Long-term medication can cause side effects to patients. Therefore, alternative treatments for anxiety disorder such as the use of probiotics as dietary supplements to improve anxiety symptoms by regulating the intestinal flora and repairing the intestinal mucosal barrier are simple, economic, and effective [52,53]. In previous studies, Yeon, Burokas, Liu, and Hsiao et al. showed that for mice anxiety caused by different methods, probiotic intervention could improve the anxiety behavior and related factor levels. This suggests that microbial modulation therapy may be an effective and safe treatment for anxiety disorders [17,54,55,56]. In the current study, results confirmed that *Lactiplantibacillus plantarum LZU-J-TSL6*, which produced high GABA, could significantly increase the GABA level in the mice hippocampus and restore the levels of related factors (5-HT, BDNF, and GFAP) disordered by restraint stress modeling. In addition, it could significantly improve the time of mice entering the center of the activity field and the residence time of independent activities, which were not different from the normal group. The results indicated that probiotic intervention had a potential effect on the improvement of anxiety symptoms in mice.

The analysis of different metabolites revealed that D-glutamine and GABA were actively metabolized after *LZU-J-TSL6* intervention, with the neuroactive ligand-receptor interaction metabolism pathway. The D-glutamine can provide energy support for intestinal cells, promote the secretion of sIgA by intestinal mucosal lymphocytes, prevent apoptosis of intestinal barrier cells, and bacterial translocation [57]. Previous studies have shown that the most important function of glutamine is to provide an energy source for rapidly proliferating cells such as intestinal mucosal epithelium, and it has a protective effect on the intestinal barrier [58,59]. The damage of the intestinal barrier is the primary mechanism by which psychological and pathological factors destroy the intestinal microecology and cause flora structural disorders. After the integrity of the intestinal barrier is compromised, the bacteria in the intestinal lumen shift, allowing exogenous endotoxin and bacterial amyloid to enter the body via the bloodstream and nervous system, triggering a systemic inflammatory response that could be one of the causes of degenerative neuroinflammation [52]. In addition, intervention with *LZU-J-TSL6* could increase the expression of the ZO-1 gene in the colon tissue of anxiety mice. As a result, *LZU-J-TSL6* can repair the intestinal mucosal barrier to inhibit the migration of intestinal bacteria and endotoxins, strengthen the tight junction between the epithelium, maintain the stability of the intestinal microecology, and regulate the immune function to relieve anxiety symptoms. 

Based on 16S rRNA high-throughput sequencing, the structural composition and abundance changes of the intestinal flora of mice in each group were evaluated at the phylum level and genus level. The number and diversity of OTUs of the mice in the anxiety model group increased. The flora analysis revealed that the number of pathogenic bacteria in the anxiety model group increased when compared to other groups, while it decreased in the LZU-J-TSL6 group and GABA group, where the number of OTUs was relatively small and their diversity declined. At the phylum level, *Bacteroidota* and *Firmicutes* are the most common bacteria. The relative abundance of *Bacteroides* was higher in the *LZU-J-TSL6* group and GABA group compared to the model group, whereas the relative abundance of *Sclerotinia* was lower in the *LZU-J-TSL6* group and GABA group. Some studies have found that Chinese medicine treatment reduces the phylum *sclerophyllum* and increases the phylum *Bacteroidota*, implying that thick-walled bacteria are related to mental diseases such as Alzheimer’s disease and autism [43,44].

At the genus level, the relative abundance of beneficial bacteria *Bacteroides* and *Muribaculum* was significantly increased in the *LZU-J-TSL6* and GABA groups. *Coriobacillae*-UCG-002 and *Dubosiella* were found to be reduced, and *Helicobacter* showed a declining trend. A study has shown that the acidification of the culture medium by *Bacteroides* spp. could induce the production of GABA [21], and the ingestion of *LZU-J-TSL6* might increase the relative abundance of *Bacteroides* spp., indicating that *LZU-J-TSL6* could enrich the original GABA-producing bacteria. While producing GABA in vivo, it further enhances the anxiety-relieving effect. A study found that Tibetan fermented milk can improve the cognitive dysfunction of Amyloid Precursor Protein (APP)/Presenilin-1 (PS1) in [APP/PS1] mice, as well as increase the intestinal microbial diversity and the relative abundance of *Muribaculum*, indicating that the cognitive function is positively correlated with the abundance of *Muribaculum* [46]. In the current study, the relative abundance of *Muribaculum* was also significantly enhanced in *LZU-J-TSL6*. Studies [47,48,49] have shown that the detection rate of anxiety and depression in the population infected with the pathogenic bacterium *Helicobacter pylori (HP)* is significantly higher than in people who are not infected with *HP*. There is a correlation between HP infection and anxiety and depression, which is explained by the mutual promotion effect demonstrated by the brain-gut axis as a bridge. After *HP* infection, anxiety and depression are more likely to occur, and anxiety and depression will have the opposite effect on the *HP*-infected population. A previous study showed that flavonoid treatment rectifies the imbalance of intestinal microbes by reducing the quantity of pathogenic *Dubosiella* bacteria [50]. The supplementation of cellulose might increase the number of colonic mucosal microbiota metastases and decrease the relative abundance of the pathogenic bacterium *Coriobacillaceae_UCG-002*, having a potential role in reducing intestinal inflammation and protecting the large intestine of mammals [51]. The ingestion of *LZU-J-TSL6* had a positive effect on the reduction of the relative abundance of the pathogenic *Helicobacter, Coriobacillae-UCG-002, and Dubosiella* bacteria. Results indicated that *Lactiplantibacillus plantarum LZU-J-TSL6* could effectively regulate the structure and diversity of intestinal flora, change the intestinal environment, and enrich the genera that actively resist anxiety.

Based on the results of this study and compared with previous studies (as shown in Table 3), it was indicated that *Lactiplantibacillus plantarum LZU-J-TSL6*, could affect the abundance of bacteria and genera related to the intestinal flora, maintain the stability of intestinal microecology, and repair the intestinal mucosal barrier. It could also increase the GABA synthesis in the mice hippocampus and improve the metabolism of related substances through exogenous pathways. 

## 5. Conclusions

In this study, a candidate probiotic strain *LZU-J-TSL6* with high GABA production was screened and its efficacy in relieving anxiety induced by restraint was also explored. The strain is intragastrically administered to mice and has been shown to effectively improve GABA content in the hippocampus, improve the metabolism level of related substances such as glutamine, repair the intestinal mucosal barrier, and regulate the structure and function of the intestinal microbial community. In comparison to pharmacological treatments, the consumption of probiotics appears to be a low-cost and low-side-effect adjuvant treatment that could be an ideal strategy for the alleviation of anxiety disorder.

## Figures and Tables

**Figure 1 foods-11-03596-f001:**
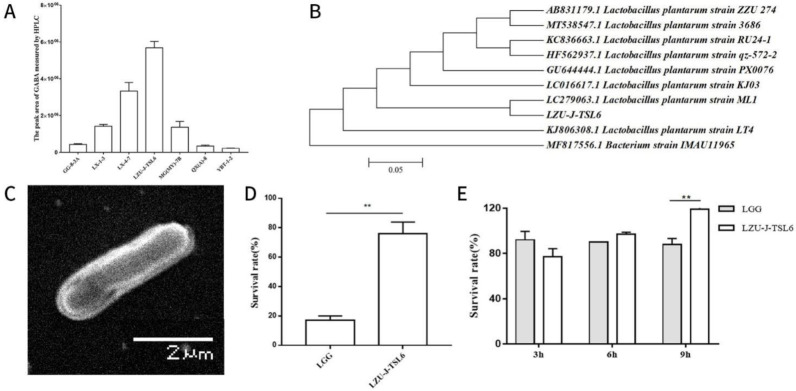
Screening and identification of the *LZU-J-TSL6* strain: Quantitative detection of γ-aminobutyric acid (GABA) production ability of different strains by high-performance liquid chromatography (**A**), Phylogenetic tree of the *LZU-J-TSL6* strain and related bacteria (**B**), Scanning electron microscopy morphology of *LZU-J-TSL6* (**C**), and the ability of the *LZU-J-TSL6* strain to resist acid (**D**) and choline (**E**) in vitro. Bars show the mean ± SD (*n* = 3 parallel trials per group). ** *p* < 0.01.

**Figure 2 foods-11-03596-f002:**
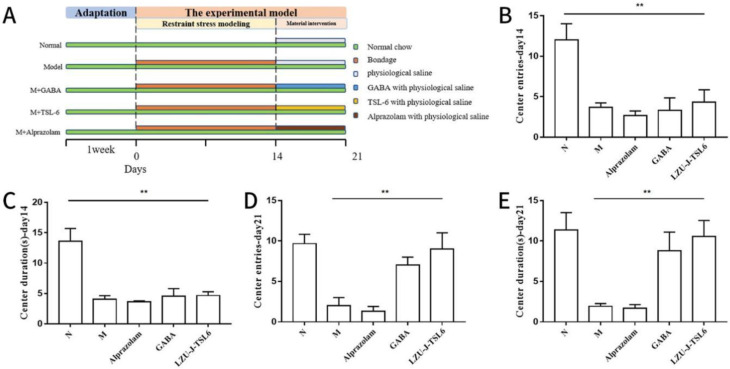
Effect of *LZU-J-TSL6* on the behavior in mice with anxiety disorders: Experimental chart of *LZU-J-TSL6* alleviating anxiety in mice (**A**). Number of entries (**B**) and time spent in the center (**C**) during open-field exploration (14 day). Number of entries (**D**) and time spent in the center (**E**) during open-field exploration (21 day). Bars show the mean ± SD (*n* = 5 mice per group) ** *p* < 0.01.

**Figure 3 foods-11-03596-f003:**
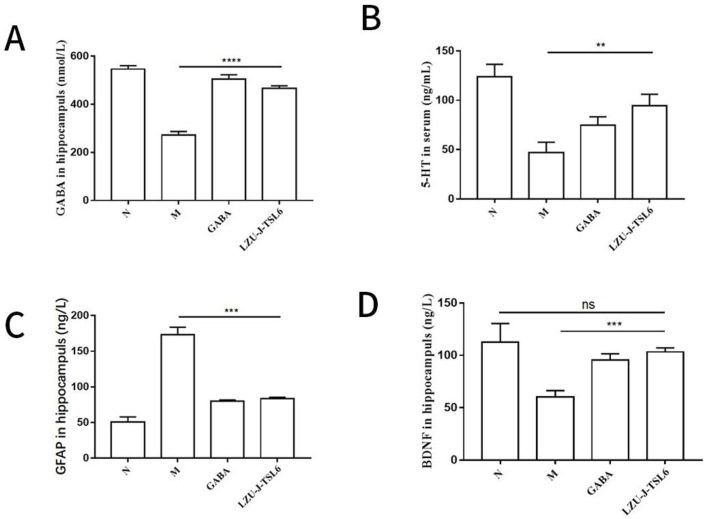
Effect of *LZU-J-TSL6* on the hippocampus and serum markers in mice with anxiety disorder: Changes in the levels of markers in the hippocampus of mice (GABA, BDNF, and GFAP) (**A**–**C**). Changes of 5-HT content in the serum of mice (**D**). Bars show the mean ± SD (*n* = 5 mice per group) ** *p* < 0.01; *** *p* < 0.001; **** *p* < 0.0001. ns: there is no significant difference between the two groups.

**Figure 4 foods-11-03596-f004:**
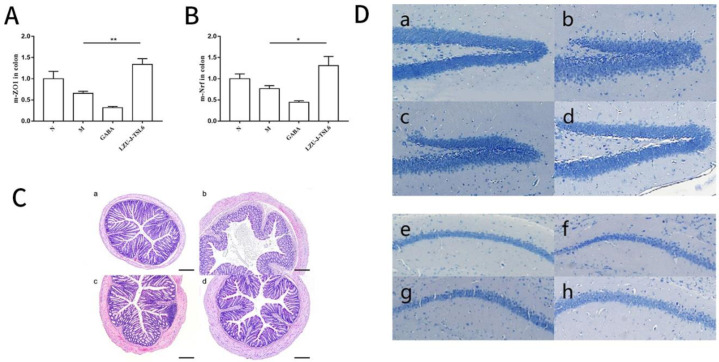
The effect of *LZU-J-TSL6* on the intestinal environment and hippocampus tissue of mice with anxiety: The expression of Nrf-2 and ZO-1 genes in the colon tissues of mice in each group (**A**,**B**). Observation of HE staining of colon tissue of mice in each group (**C**). ((**a**). Control group; (**b**). Model group; (**c**). GABA group; (**d**). *LZU-J-TSL6* group). Observation of Nissl body staining in the DG and CA1 regions of the hippocampus of mice in each group (**D**). ((**a**,**e**): Control group; (**b**,**f**): Model group; (**c**,**g**): GABA group; and (**d**,**h**): *LZU-J-TSL6* group). Bars show the mean ± SD (*n* = 5 mice per group) * *p* < 0.05; ** *p* < 0.01.

**Figure 5 foods-11-03596-f005:**
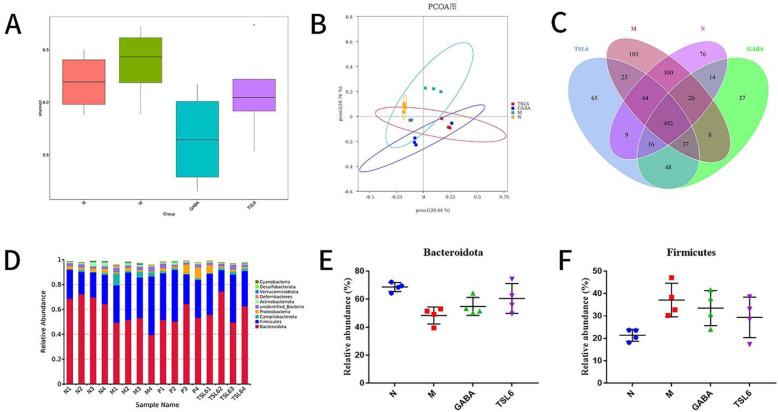
*LZU*−*J*−*TSL6* regulates model-induced intestinal microbiota disorders in mice with anxiety disorders: Box plot of differences in Shannon index between groups (**A**). PCoA analysis based on Unweighted Unifrac distance (**B**). A Venn diagram analysis of OTU overlap between different microbiomes (**C**). Comparison of the relative abundance of phylum between different groups (**D**). Relative abundance of *Bacteroidota* and *Firmicutes* (**E**,**F**). The bars indicate the mean ± SD (*n* = 4 mice/group).

**Figure 6 foods-11-03596-f006:**
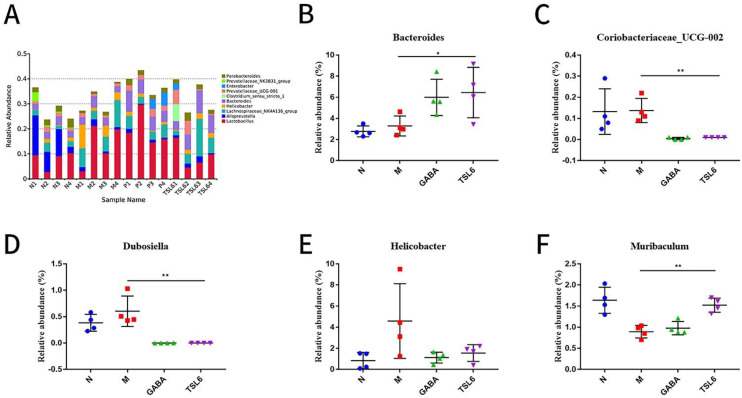
Regulation of the level of the intestinal microbiota of mice in each group of *LZU*−*J*−*TSL6*: Comparison of the relative abundances of phylum between different groups (**A**). Relative abundances of the bacterial genera with related changes (*Bacteroides*, *Coriobacteriaceae−UCG−002*, *Dubosiella*, *Helicobacter*, and *Muribaculum*) (**B**–**F**). The bars indicate the mean ± SD (*n* = 4 mice/group) * *p* < 0.05; ** *p* < 0.01.

**Figure 7 foods-11-03596-f007:**
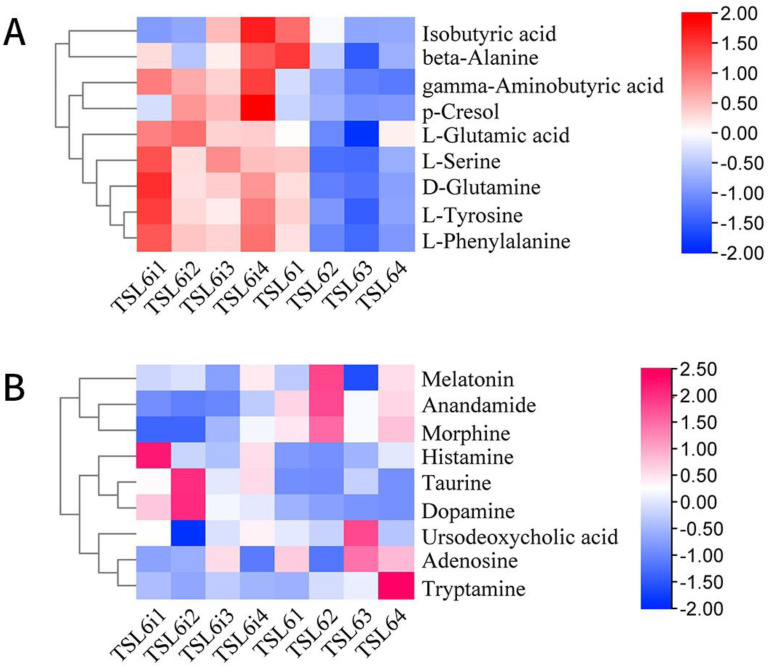
Analysis of differential metabolites in mice before and after intervention with *LZU−J−TSL6***.** Chromatographic cluster analysis of differential metabolites before (**A**) and (**B**) after *LZU−J−TSL6* intervention (a negative ion mode, b positive ion mode).

**Figure 8 foods-11-03596-f008:**
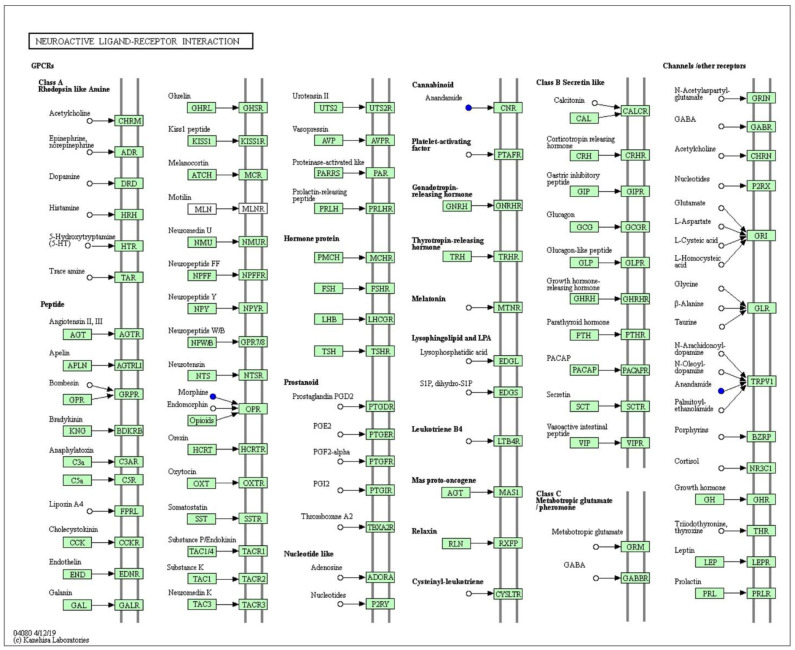
Prediction of metabolic pathway in mice before and after intervention with *LZU-J-TSL6*. The KEGG metabolic pathway analysis diagram–metabolic pathway of neuroactive ligand-receptor interaction.

**Table 1 foods-11-03596-t001:** Fermented food sample information.

Number	Sample Origin	Number of Samples	Production Method	Strain Named
1	Guoluo (Maqin), Qinghai, China	1	Herders homemade	GL
2	Guoluo (Maqin), Qinghai, China	1	commercial product	GL(AS)
3	Guoluo (Maqin), Qinghai, China	1	commercial product	GL(GQ)
4	Yushu (Jiegu), Qinghai, China	1	Herders homemade	YJG
5	Yushu (Nangqian), Qinghai, China	1	Herders homemade	YNQ
6	Yushu (Chengduo), Qinghai, China	1	Herders homemade	YCD
7	Yushu (Batang), Qinghai, China	1	Herders homemade	YBT
8	Tongren (Ningxiu), Qinghai, China	1	Herders homemade	NX
9	Tongren (Guanxiu), Qinghai, China	1	Herders homemade	GX
10	Tongren (Duowa), Qinghai, China	1	Herders homemade	DW
11	Tongren (Guashenzi), Qinghai, China	1	Herders homemade	GZ
12	Tongren (Suonaihe), Qinghai, China	1	Herders homemade	SN
13	Tongren (Duofudun), Qinghai, China	1	Herders homemade	DFD
14	Linxia, Gansu, China	4	Farmer homemade	LX
15	Yuzhong, Gansu, China	1	Farmer homemade	YZ
16	Qin’an, Gansu Province, China	10	Farmer homemade	QA
17	Gangu, Gansu, China	10	Farmer homemade	GG
18	Buy different brands and sources of Jiangshui in Lanzhou vegetable market	13	commercial product	QX(A), QX(AY), WD(Y), WD(L), JF(J), MG(M), QY(QT), S(L), TS(Q), TS(W), TS, JS(A), MG(MY)

Name that screened strain with the initial of the local name of the sample.

**Table 2 foods-11-03596-t002:** Detailed information on bacterial and fungal primers used in this study.

Name	Pre-Primer Sequence	Pre-Primer Sequence
Bacterium 16S rRNA (V3 + V4)	5′-ACTCCTACGGGAGGCAGCA-3′	5′-GGACTACHVGGGTWTCTAAT-3′
Fungus ITS1 region	5′-CTTGGTCATTTAGAGGAAGTAA-3′	5′-GCTGCGTTCATCGATGC-3′

**Table 3 foods-11-03596-t003:** Comparison of obtained results in anxiety upon the use of *Lactiplantibacillus plantarum LZU-J-TSL6* and probiotics reported in previous studies.

Name	Source	Properties	Function	References
*Lactobacillus plantarum LZU-J-TSL6*	Jiangshui	High yield of GABA	Regulating intestinal flora diversity. Produce glutamine. Repair intestinal mucosal barrier. Increase GABA content in the hippocampus. Repair the levels of BDNF and GFAP in hippocampus of mice. Reduce the damage of Nissl corpuscles in hippocampus. Alleviate anxiety symptoms in mice.	This study
*Lactobacillus rhamnosus HN001*	/	/	Reduce the incidence of postpartum depression or anxiety	[60]
*Lactobacillus helveticus NS8*	Natural fermented dairy	/	Improve cognitive dysfunction. Reduce the levels of plasma corticosterone and adrenocorticotropic hormone. Adjust the level of inflammatory factors. Restore the levels of 5-HT and norepinephrine in hippocampus. Increase the expression of BDNFmRNA in hippocampus.	[61]
*Bifidobacterium adolescentis IM38*	Human faecal microbiota	/	Significantly increase the time for mice to enter the bilateral open arms. Decreased levels of blood corticosterone and IL-6 in mice with or without fixed stress. Reduce the level of TNF-α in blood of stressed mice. IM38 can alleviate anxiety by regulating benzodiazepine sites on GABA receptor and regulate the expression of stress-related cytokines.	[62]

## Data Availability

Data available on request from the authors.

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
