# Peer review of "Overcoming Anxiety Disorder by Probiotic Lactiplantibacillus plantarum LZU-J-TSL6 through Regulating Intestinal Homeostasis"

_foods, 2022, doi:10.3390/foods11223596_

Round 1

Reviewer 1 Report

Regarding the manuscript entitled ‘’ Overcoming anxiety disorder by probiotic Lactobacillus planta- 2 rum LZU-J-TSL6 through regulating intestinal homeostasis’’

L19. Model group or do you mean control

L20 missing p value here

L26-27. Please rephrase

L29.oxytocin, GABA, 5-HT, norepinephrine, dopamine and acetylcholine, please mention these results in the results of abstract

L34-37. Add ref

L41-42. Please rephrase

L99. Hypothesis is missing

L125. Double-check the subheading

L126. Add more details about 16S rRNA gene sequencing of the strain

L152-153. ‘’ During the experiment, the mice were randomly 152 divided into five groups, and six mice per cage were placed under the same condition for 153 the experiment’’ What is the number of replicates, and how the authors statistically analyzed the obtained data?

The design of the experiment is a completely randomized design or completely randomized block design. Please clarify

L165. on which basis the authors chose the dose?

L199. Add ref.

Conclusion, please be precise with the recommendation to the readers.

Author Response

On behalf of my co-authors, we are very grateful to you for giving us an opportunity to revise our manuscript. We appreciate you very much for your positive and constructive comments and suggestions on our manuscript entitled “Overcoming anxiety disorder by probiotic Lactiplantibacillus plantarum LZU-J-TSL6 through regulating intestinal homeostasis” (Manuscript ID: foods-1963443).

We have studied the reviewers’ comments carefully and revised our manuscript according to the comments. The following are the responses and revisions I have made in response to the suggestions on an item-by-item basis. Thanks again to the hard work of the editor and reviewer!

Point 1: L19. Model group or do you mean control.

Response 1: Dear reviewer, MODELS here does not mean model group in grouping. It has the same meaning as MODEL in model creatures. If there is any ambiguity in this way, we can consider putting it another way.

Point 2: L20 missing p value here.

Response 2: The P values have been supplemented and highlighted in green.

Point 3: L26-27. Please rephrase

Response 3: The sentence was rewritten and highlighted in green.

Point 4: L29.oxytocin, GABA, 5-HT, norepinephrine, dopamine and acetylcholine, please mention these results in the results of abstract.

Response 4: Dear reviewer, the factors such as oxytocin, dopamine and norepinephrine mentioned in the introduction of the article are to show that the intake of probiotics by the body will indeed produce substances that can affect the nervous system, thus providing a feasible basis for the experiment. Only the results of GABA, BDNF, GFAP and 5-HT were obtained, so we don't think it is necessary to mention oxytocin in the abstract.We have also corrected this section and highlighted it in green.

Point 5: L34-37. Add ref.

Response 5: Dear reviewer, it is not necessary to cite references in our article lines 34-37. This is between the last sentence of the abstract and the title of the introduction.

Point 6: L41-42. Please rephrase.

Response 6: The sentence was rewritten and highlighted in green.

Point 7: L99. Hypothesis is missing.

Response 7: Thanks to Reviewer for the reminder,We have corrected this sentence and highlighted it in green.

Point 8: L125. Double-check the subheading.

Response 8: Dear reviewer, there is no subtitle here. However, thank you for reminding me that there is something wrong with the subtitle of 2.2. We have made corrections and highlighted it in green.

Point 9: L126. Add more details about 16S rRNA gene sequencing of the strain.

Response 9: Dear reviewer, we have made some additions here, but the whole gene sequencing process of the strain is done by the company, we just cultivate the strain and send the sample to the company.

Point 10: L152-153. “During the experiment, the mice were randomly 152 divided into five groups, and six mice per cage were placed under the same condition for 153 the experiment” What is the number of replicates, and how the authors statistically analyzed the obtained data?

The design of the experiment is a completely randomized design or completely randomized block design. Please clarify.

Response 10: Dear reviewer, we didn't find the problem of grouping mice here, so we felt a little confused. But we explained the problem of mice grouping: At the start of the experiment, mice (n=35) were randomly divided into 5 groups (n=7 per group), but during the modeling period of the experiment, mice died at a different rate in each group. As a result, at the end of the experiment, the number of mice was 6 in the N group, 4 in the M group, 4 in the GABA group, 5 in the TSL6 group, and 4 in the alprazolam group, respectively. In order to keep the replicates number equal in all groups, we used 4 mice (n=4) in each group for final data analysis. So, the number of replicates number is 4. The data were analyzed with GraphPad Prism software (Version 7.00) and with Mean and SD.The study design is completely randomized.

Point 11: L165. on which basis the authors chose the dose?

Response 11: Dear reviewer, we have indicated where appropriate and highlighted in green.The GABA dose of mice was calculated from the amount of bacterial solution administered by gavage combined with the results obtained by liquid chromatography above. The dose of alprazolam used in mice was calculated by the human-animal dose conversion formula based on the maximum daily dose taken by humans.

Point 12: L199. Add ref.

Response 12: We have added references and highlighted them in green.

Reviewer 2 Report

Dear Editors and authors,

Major comments

1-In the culture media, the compound γ-aminobutyric acid (GABA) is produced in very small amounts. Why did the researcher not use an incentive in the production medium such as sodium glutamate?

Minor comments

1- The names of the bacteria should be written in italics throughout the manuscript.

2-The names of lactic acid bacteria should be written according to the modern nomenclature that has been used since 2020 such as Lactobacillus plantarum correcting to Lactiplantibacillus plantarum, throughout the manuscript.

3- Many methods require scientific references. It is not permissible to write the method without a scientific reference, such as XCV, Scanning electron microscope, and .........etc. 

4- I suggest you to add (Niamah, A. (2019). Ultrasound treatment (low frequency) effects on probiotic bacteria growth in fermented milk. Future of Food: Journal on Food, Agriculture and Society7(2), Nr-103. for growth isolates method page 3 line 108. and 

Kaláb, M., Yang, A. F., & Chabot, D. (2008). Conventional scanning electron microscopy of bacteria. Infocus magazine10, 42-61. for Scanning electron microscope of bacteria shapes method.

‏4- The PCR program was not mentioned in the 16S rRNA test and the gastrointestinal bacteria diagnosis test? Why should the program be mentioned?

5- Correct  line 125 page 3.

6-Page 3 line 102, the author did not mention what are the sources of isolates(288)? And where did you get isolated  ?

7-Figure 7 shows the compounds produced by bacteria, but in the working methods the author did not mention the method.

8- Suggest adding a table showing the amount of produced from γ-aminobutyric acid (GABA) using the bacterial seven isolates.

9-In the abstract of the manuscript, the researcher did not mention the amount of output from GABA.

Author Response

On behalf of my co-authors, we are very grateful to you for giving us an opportunity to revise our manuscript. We appreciate you very much for your positive and constructive comments and suggestions on our manuscript entitled “Overcoming anxiety disorder by probiotic Lactiplantibacillus plantarum LZU-J-TSL6 through regulating intestinal homeostasis” (Manuscript ID: foods-1963443).

We have studied the reviewers’ comments carefully and revised our manuscript according to the comments. The following are the responses and revisions I have made in response to the suggestions on an item-by-item basis. Thanks again to the hard work of the editor and reviewer!

Major comments

Point 1: In the culture media, the compound γ-aminobutyric acid (GABA) is produced in very small amounts. Why did the researcher not use an incentive in the production medium such as sodium glutamate?

Response 1: If stimulants are added to the culture medium, GABA is bound to be produced during the culture of the bacterial solution used in the experiment, then the test mice will be supplemented with additional GABA during the gavage of the bacterial solution. The aim of the present study was to investigate the positive effects of GABA-producing strain on mice. In addition, if stimulants were added to the medium, it may not be better stated that the use of this strain alone is more beneficial than GABA supplementation.

Minor comments

Point 1: The names of the bacteria should be written in italics throughout the manuscript.

Response 1: We modified this and highlighted it in red.

Point 2: The names of lactic acid bacteria should be written according to the modern nomenclature that has been used since 2020 such as Lactobacillus plantarumcorrecting to Lactiplantibacillus plantarum, throughout the manuscript.

Response 2: We modified this and highlighted it in red.

Point 3: Many methods require scientific references. It is not permissible to write the method without a scientific reference, such as XCV, Scanning electron microscope, and .........etc. 

Response 3: Thanks to the reviewers for their suggestions, we have added references .We also added references where necessary.

Point 4: The PCR program was not mentioned in the 16S rRNA test and the gastrointestinal bacteria diagnosis test? Why should the program be mentioned?

Response 4: Dear reviewer, thank you for your reminder. We have supplemented it in the corresponding places.

Point 5: Correct line 125 page 3.

Response 5: Dear reviewer, the inspection here has not found anything that needs to be corrected. Please point out more specific problems, and we are very willing to make changes.

Point 6: Page 3 line 102, the author did not mention what are the sources of isolates(288)? And where did you get isolated ?

Response 6: All strains were isolated from fermented food-Jiangshui.This is illustrated on page 3 of the manuscript at line 105 and highlighted in red.

Point 7: Figure 7 shows the compounds produced by bacteria, but in the working methods the author did not mention the method.

Response 7: Dear reviewer, thank you for your reminder. We have supplemented it in the corresponding places and highlighted it in red.

Point 8: Suggest adding a table showing the amount of produced from γ-aminobutyric acid (GABA) using the bacterial seven isolates.

Response 8: Dear reviewers, thank you very much for your suggestions. We thought that the column diagram based on the peak area could intuitively reflect the GABA production ability of each strain. Therefore, we do not think it is necessary to make forms. Of course, it is necessary to confirm the yield, so we explained the yield of LZU-J-TSL6 in the paper(Page six, line 260). If you insist on making the form as necessary, we will take your comments and make further modifications.

Point 9: In the abstract of the manuscript, the researcher did not mention the amount of output from GABA.

Response 9: Thanks to Reviewer for the reminder,we've added and highlighted it in red.

Round 2

Reviewer 1 Report

Dear authors

Thank you for your revisions 

Author Response

Dear reviewer, thank you very much for your approval of our reply, and we thank reviewer -2 for the second round of comments.

Reviewer 2 Report

Dear Editors, 

1-The author did not answer the main questions of the evaluation

2- Many scientific names are still written in italics see the title of manuscript.

3-The author did not indicate the source of the isolates used in the manuscript.

4-No program was added to the 16S rRNA test.

5-The author had to provide answers to the questions submitted by the evaluator and review the manuscript to be acceptable for publication and not to ignore the evaluator’s questions.

Author Response

We thank the editor for their services to publish our work in peer review “Journal of Foods”. In response to the editor report we have made all the suggested revision in the manuscript revised file according to the reviewers comments, and hope that our response will meet the requirement of the comments received from the editor and reviewers. Please find below a detailed point-by-point response to all comments (reviewers’ comments in black, our replies in red). The changes made in the manuscript in light of the reviewer's comments have been highlighted in red in the revised manuscript file accordingly.

Point 1: The author did not answer the main questions of the evaluation

Response 1: In an in vitro experiment, we added sodium glutamate to the culture medium used for screening strains with GABA production ability so that TSL6 was found to have a stronger ability to produce GABA. Therefore, this strain was selected for subsequent experiments.

However, for the in vivo mouse experiments, we did not include sodium glutamate in the strain medium. We believe that if sodium glutamate is added, the unused sodium glutamate and the transformed GABA of the strain will be given to the mice together during the gavage treatment of the bacterial solution. In this process, the amount of sodium glutamate and GABA is uncertain, which will have an impact on the experimental results. In addition, if sodium glutamate is intragastrically administered to mice at the same time as a bacterial solution, the flora in the mice will also use sodium glutamate which may will transform into other products, which will further introduce uncertainty to the experiment results. The aim of this study was to investigate whether the intake of the probiotic lactobacillus strain could regulate the gut microbiota, repair the intestinal barrier, and alleviate anxiety symptoms in mice. If sodium glutamate was added to the medium and GABA was additionally supplemented, the role played by the strains would not be adequately illustrated.

Although the raised concern is quite worthy, however, it will need a different experimental design where to set animal groups; sodium glutamate with probiotic strains and sodium glutamate without probiotic strains, and compare them, to obtain further experimental results and conclusions, which we can consider in future experiments. But here in this experiment, our aim was only to evaluate the probiotic Lactobacillus strains. 

Point 2: Many scientific names are still written in italics see the title of manuscript.

Response 2: Dear reviewer, we rechecked the entire manuscript and italicized the scientific names accordingly, and highlighted it in red.

Point 3: The author did not indicate the source of the isolates used in the manuscript.

Response 3: All strains were isolated from fermented food (Jiangshui), a well-known Chinese traditional food. The details are already illustrated on page 3 of the manuscript in lines 94-102, highlighted in red. In addition, we have also tabulated the sources of the isolates, see Table I for details. Meanwhile, we have referenced the Table 1 detail in the manuscript section 2.1, and highlighted it in red.

Point 4: No program was added to the 16S rRNA test.

Response 4: The program details are added to the 16S rRNA test, manuscript section 2.2、2.10.1.and 2.10.2, and revised the text accordingly. The revision is as follows:

2.2 16S rRNA gene sequencing and phylogenetic analysis of strain LZU-J-TSL6

The bacterial suspension from the culture of strain LSU-J-TSL6 was submitted to the company as a sample for the subsequent sequencing process.TIANamp Bacteria DNA Kit (TIANGEN Biotechnology, China) was used to obtain high quality genomic DNA of strains. The 16S rDNA was amplified as described by Sibley et al.[30]The 16S rRNA gene sequencing of the strain was performed by Beijing qingke biotechnology co. Ltd. and identified by BLAST engine (NCBI)[31]. The phylogenetic tree was made by MEGA (molecular evolutionary genetic analysis, version 6.0) software[32]. The strain LSU-J-TSL6 nucleic acid sequence is shown in supplementary table 5

2.10.1. Extraction and PCR amplification of genomic DNA from intestinal microbiota

The genomic DNA of the fecal samples was extracted according to the instructions using the DNA strong extraction kit produced by Bio Corp. in the United States, and then the purity and concentration of DNA were detected by agarose gel electrophoresis. An appropriate amount of sample DNA was taken in a centrifuge tube and the samples were diluted to 1ng/μl with sterile water. According to the selection of sequencing region, PCR was performed using the diluted genomic DNA as a template using a specific primer with Barcode. The entire process uses Phusion High-Fidelity PCR Mastermix with GC Buffer from New England Biolabs and high-performance hi-fi enzymes to ensure amplification efficiency and accuracy[37]. The PCR expansion and high-throughput sequencing of the samples, and the amplification primers designed with the variable sequence of the 16SrRNA V3-V4 region as the target are shown in Table 2.

2.10.2. Mixing and purification of PCR products

The PCR mixture (25 μl) contained 1x PCR buffer, 1.5 mM magnesium chloride, 0.4 micron concentration of deoxynucleoside triphosphate, 1.0 micron concentration of each primer and 0.5 U Ex Taq(TaKaRa, Dalian), and 10 ng of soil genomic DNA. The PCR amplification protocol consisted of an initial denaturation at 94 ℃ for 3 min followed by 30 cycles of 94 ℃ for 40s, 56 ℃ for 60s, 72 ℃ for 60s, and a final extension at 72 ℃ for 10 min. Two PCR reactions were performed for each sample and combined after PCR amplification[38].DNA concentration and purity were determined by NanoDrop (Thermo Scientific, USA). PCR products were examined by electrophoresis on a 1% agarose gel. The product was recovered from the target band using a gel recovery kit provided by qiagen, and the concentration and quality of the product were determined by Nanodrop.

After the library passed the test, sequencing and computational analysis was performed using the Illumina NovaSeq platform (NovaSeq6000) with the NovaSeq Reagent Kit V3-V4 at (Shanghai Baiqu Biomedical Technology Co., Ltd) Shanghai, China, according to the standard protocols.

Point 5: The author had to provide answers to the questions submitted by the evaluator and review the manuscript to be acceptable for publication and not to ignore the evaluator’s questions.

Response 5: We have addressed all the concerns raised by the reviewers with appropriate answers and revised the manuscript accordingly. The revision is marked in red.
